# Neonatal Airway Abnormalities

**DOI:** 10.3390/children9070944

**Published:** 2022-06-24

**Authors:** Adithya Srikanthan, Samantha Scott, Vilok Desai, Lara Reichert

**Affiliations:** 1Albany Medical College, Albany, NY 12208, USA; srikana@amc.edu (A.S.); scotts6@amc.edu (S.S.); desaiv1@amc.edu (V.D.); 2Department of Otolaryngology, Albany Medical Center, Albany, NY 12208, USA

**Keywords:** neonate, airway obstruction, respiratory tract, congenital neck mass, surgery, trachea, larynx

## Abstract

Neonatal airway abnormalities are commonly encountered by the neonatologist, general pediatrician, maternal fetal medicine specialist, and otolaryngologist. This review article discusses common and rare anomalies that may be encountered, along with discussion of embryology, workup, and treatment. This article aims to provide a broad overview of neonatal airway anomalies to arm those caring for these children with a broad differential diagnosis and basic knowledge of how to manage basic and complex presentations.

## 1. Introduction

The neonatal airway is made up of complex anatomy with equally complex function. The upper airway extends from the oral cavity to the trachea, providing a channel for respiration, but also serving as a means of phonation and a path for clearing respiratory secretions. Further, the vocal cords serve as a protective structure for the lower airway, working with the epiglottis to prevent secretions and ingested food from entering the trachea. As the airway serves as the sole conduit for respiration, compromise of this tract can lead to potentially life-threatening conditions. A variety of congenital interruptions, craniofacial abnormalities, and surgical interventions affecting any part of the airway early in life can predispose a neonate to such compromise. As many of these pathologies can present with similar symptoms or no immediate symptoms at all, a high degree of suspicion is required to identify them as early as possible and intervene where necessary. Here, we delineate some common neonatal airway abnormalities, along with presentation, pathophysiology, and management.

## 2. Literature Review

### 2.1. Laryngomalacia

Laryngomalacia is an anomaly of the larynx commonly seen in neonates that leads to an inward collapse of supraglottic airway tissue upon inspiration [1]. It is the most common cause of stridor in neonates [2]. Along with stridor, a neonate with laryngomalacia can demonstrate poor feeding, gagging, respiratory distress, regurgitation, and failure to thrive [3,4]. In severe cases, pectus excavatum, right heart strain, or cor pulmonale may be present.

Laryngomalacia affects the structures above the true vocal cords, the supraglottis, and is theorized to occur secondary to abnormal embryologic development of the laryngeal apparatus, although the true etiology is not known. Along with the anatomic and cartilaginous defects, neurologic defects may contribute to laryngomalacia. This theory proposes that weak laryngeal tone leads to laryngeal collapse and laryngomalacia [5]. At 4 weeks gestation, the larynx first develops in the embryo, and by 8 weeks all major cartilage structures are present [6]. There are several defects noted to contribute to congenital laryngomalacia, including redundant supraglottic mucosa, an elongated epiglottis that collapses inward and posteriorly (the classic “omega-shape”), and foreshortened aryepiglottic folds seen in Figure 1 [5].

Physicians should be aware of the clinical presentation of an infant with laryngomalacia as its frequency is as high as 1 in 2000 the general population [3]. Early diagnosis is essential, as laryngomalacia can cause feeding difficulties and failure to thrive, affecting a neonate’s long-term quality of life and development [3]. A complete history and physical exam focusing on the oral cavity, nose, and neck should be performed in any infant with suspected laryngomalacia, with referral to an otolaryngologist for laryngoscopy. Common pathologies that should be excluded include cleft lip and palate, stenosis of the choana and pyriform aperture, micrognathia, subglottic stenosis, and vocal cord paralysis [3,4]. In 90% of neonates, symptoms of laryngomalacia resolve on their own [7]. Some benefit from reflux management. In patients with severe respiratory symptoms, supraglottoplasty can be curative [7]. Various techniques have been described for supraglottoplasty, including cold steel and laser procedures. Radiofrequency ablation of redundant and obstructive laryngomalacia has been described as a newer technique to treat laryngomalacia [8].

### 2.2. Vocal Cord Paralysis

Vocal cord paralysis (VCP) is a common airway abnormality in neonates in which one or both of the vocal cords are unable to adduct or abduct properly [9]. It comprises up to 10% of all congenital laryngeal lesions, second only to laryngomalacia, and has been found in 3.5% of preterm infants in the Neonatal ICU, with incidence increasing to 18% in patients born before 26 weeks GA [10,11].

VCP most often arises from disruption of the recurrent laryngeal nerves (RLN), which innervate the intrinsic muscles of the larynx, and paralysis can result from pathologies anywhere along the RLN path [9,12]. Congenital causes of RLN damage include Arnold–Chiari malformation, cranial mass, and hydrocephalus, transposition of the great vessels, aortic coarctation, and mediastinal masses, with mediastinal pathologies conferring a greater risk to the left RLN [13]. VCP is most often iatrogenic, with 18% to 69% of attributable to procedures [9]. Cardiothoracic surgery imparts the highest risk, with postoperative incidence as high as 59% and 20% after aortic arch or ductus arteriosus procedures, respectively [14,15,16,17,18]. VCP may also result from birth trauma, TEF/esophageal atresia repairs, and chemotherapy-induced neuropathy, with mechanical interruptions least likely to return to normal function [9,19,20,21]. Idiopathic and neurologic paralysis, comprising up to 35% and 16% of patients, has the highest spontaneous return to normal function, with 64% of idiopathic patients regaining normal function within 4 years and 71% of neurologic patients within 2 years [22]. Unilateral patients may also experience quicker return to function due to compensation by the unaffected cord [9].

Initial presentation can differ significantly between unilateral and bilateral VCP, and diagnosis is made through laryngoscopy. Bilateral VCP has a higher incidence of severe respiratory manifestations, with retractions, apnea, and the need for respiratory support, [22,23,24] whereas unilateral VCP presents with dysphagia and dysphonia, or a breathy cry (Figure 2) [9]. As dysphagia is a common symptom of VCP, a thorough feeding and video swallow evaluation may be warranted [9].

Bilateral paralysis often requires tracheostomy, particularly in patients with comorbid airway pathologies [25]. BVCP patients can undergo posterior cricoid split, partial cordotomy, suture lateralizations, and cricothyroid botulinum injections, often facilitating decannulation later in life [25,26,27,28]. Unilateral paralysis does not often require tracheostomy, and can be managed conservatively with measures such as side-lying [29]. Injection laryngoplasty can also be performed in older children to improve feeding, voicing, and prevent aspiration [30]. A 2020 study of pediatric otolaryngologists found only 31% of respondents performed injection laryngoplasty in patients under 12 months [31]. Laryngeal reinnervation surgery is a newer technique requiring open surgery with microscopic nerve dissection and approximation aimed at restoring function to the larynx by utilizing alternative nerves other than the recurrent laryngeal nerve or vagus nerve to activate affected laryngeal muscles [32].

### 2.3. Subglottic Stenosis

Subglottic stenosis (SGS), narrowing of the upper airway between the vocal cords and proximal trachea, occurs most commonly as an acquired defect due to prolonged intubation. Congenital subglottic stenosis is much less frequent, and its mechanism is found in neonatal embryology. The cricoid cartilage develops from the sixth pharyngeal arch. In the third month of gestation, the lumen of the larynx recanalizes. In congenital SGS, this process is abnormal, resulting in a restricted airway (Figure 3) [33].

The subglottis is the narrowest section of the neonatal airway. A normal diameter of the subglottis for a full-term neonate is 4 mm, and a subglottic diameter less than 3.5 mm is considered stenotic. Pre-term infants have a narrower subglottic diameter. Considering the already narrow airways of both full and pre-term infants, stenosis can significantly impede airflow leading to respiratory distress. Management of patients with SGS depends on the severity of the impediment. The Cotton–Myer system is a standardized way to grade the neonatal airway. In patients with Cotton–Myer Grade 1, less than 50% of the airway is stenosed. Those with grade 2 stenosis have 51–70% airway narrowing. These patients are most likely to present with mild airway symptoms such as persistent stridor and recurrent croup, and they can be managed conservatively or with endoscopic procedures including balloon dilation. Higher Cotton–Myer grades are most likely to need swift surgical management to secure a stable airway. Grade 3 denotes 71–99% stenosis, and patients are typically symptomatic. Grade 4 stenosis denotes > 99% obstruction and necessitates tracheostomy. Open reconstructive procedures, such as laryngotracheal reconstruction with cartilage grafting, may be necessary to expand the airway of patients with high-grade stenoses. Slide tracheoplasty has been discussed in very young infants as a single-stage procedure to treat stenosis without the need for temporary tracheostomy, but series are limited [34].

Some syndromes are associated with congenital subglottic stenosis. Pallister–Killian syndrome is a rare chromosomal aneuploidy of 12p that results in multiple systemic abnormalities, including progressive subglottic stenosis. In fact, most cases of congenital SGS are not isolated but are found as a symptom within a syndrome. Trisomy 21, CHARGE, and 22q11 are examples of syndromes in which SGS can occur [35,36,37].

### 2.4. Laryngeal Webs

Congenital laryngeal webs are a rare form of partial laryngeal atresia that occurs due to failure of the larynx to fully recanalize during the 9th week of embryonic development [38,39]. Laryngeal webs account for 5% of laryngeal anomalies and have an incidence of 1 in 10,000 births [40]. Laryngeal webs can lead to stridor in the first months of life, although patients diagnosed later than the neonatal period have been reported [40,41]. Other presentations of laryngeal web include dysphonia presenting as a weak cry, early laryngitis, and respiratory distress [42]. Infants with congenital laryngeal webs should be evaluated for chromosomal and cardiovascular abnormalities as chromosome 22q11 deletion syndromes have an increased incidence of laryngeal web [43].

Laryngeal webs are classified into four subtypes [44]. Type I and type II subtypes present with less than 50% glottic involvement and are milder phenotypes [44]. Type III webs have 50–75% glottic involvement, and type IV webs have 75–90% glottic involvement [44]. Surgical management varies with the severity, and types III and IV webs may require tracheostomy during the first few days or months of life [42].

### 2.5. Subglottic Hemangioma

Hemangiomas are benign and are the most common tumors of infancy, which usually appear within the first 6 months of life [45]. Most hemangiomas are cutaneous. Subglottic hemangiomas are rare, accounting for 1.5% of congenital laryngeal anomalies [45]. Subglottic hemangioma may pose a danger to airway integrity if tumor growth threatens the patency of the airway (Figure 4).

Presenting symptoms include biphasic stridor, respiratory distress, and difficulty feeding [45,46,47]. Importantly, the infant’s cry will be normal and swallowing will be unaffected unless there is an additional airway lesion [47]. Patients are often misdiagnosed with croup, and recurrent croup diagnoses that do not resolve should raise suspicion for subglottic hemangioma [47]. Additionally, patients with subglottic hemangioma often present with cutaneous hemangiomas in a “beard-like” distribution, with up to 50% of the patients with such head and neck hemangiomas having concomitant airway hemangioma [47].

Treatment of subglottic hemangioma depends on the severity of the airway obstruction. Steroids, B-blockers including propranolol, endoscopic laser resection, tracheostomy, and open excision are treatments for subglottic hemangioma [45,46,47]. If the airway is not compromised, hemangiomas can be treated with a “watch and wait” approach as they spontaneously regress over years.

### 2.6. Tracheobronchomalacia

Tracheobronchomalacia (TBM) describes airway collapse in the trachea and mainstem bronchi (Figure 5) [48]. It is a general term that encompasses several entities, including unilateral airway compression due to innominate artery compression, circumferential airway compression, and cartilage weakness leading to collapse of the airway [49]. TBM is classified into primary and secondary TBM, with primary TBM occurring in otherwise healthy infants and secondary TBM occurring in relationship to another anomaly. Tracheoesophageal fistula is nearly always associated with TBM, as it creates a weakness in a segment of the trachea. Vascular anomalies may cause TBM when there is extrinsic compression on the airway [49,50]. TBM is diagnosed when there is a significant reduction in luminal or cross-sectional diameter of the trachea or bronchi [51]. When multiple segments of the airway are not involved, the condition may be classified as tracheomalacia or bronchomalacia, rather than combined TBM.

The incidence of TBM is estimated to be 1 per 1445 infants, although this may be an underestimation [51]. There is a male predominance, and multiple studies describe an increased incidence of TBM in premature infants [48,50]. Prematurity may be a risk factor for TBM due to more compliant tracheal rings and lower tracheal muscle tone [50].

Infants with TBM present with nonspecific symptoms such as stridor, wheezing, coughing, cyanosis, breath-holding spells, and difficulty feeding [18,19,21]. TBM can be misdiagnosed as difficulty to control asthma, and other causes of stridor and cough must be ruled out such as laryngomalacia, foreign body aspiration, or upper respiratory infection [21]. Mild TBM is self-limiting and resolves as an infant grows and tracheal cartilage strengthens. CPAP, chest physiotherapy, treatment of underlying GERD, and mucolytic agents have been proposed to treat mild-to-intermediate TBM. Positive airway pressure, sometimes necessitating tracheostomy, is recommended for infants whose airway patency cannot be adequately supported with other methods. Aortopexy can be used to decrease anterior tracheomalacia in certain patients and may be performed simultaneously with repair of existing esophageal anomalies such as tracheoesophageal fistula [52]. Specialized institutions offer surgical correction of TBM by elevating the aorta away from the trachea, suturing the anterior tracheal wall to the sternum, or suturing the posterior tracheal wall to the prevertebral fascia [53,54].

### 2.7. Congenital Tracheal Stenosis/Vascular Rings

Congenital tracheal stenosis is a rare disorder defined as a narrowing of the tracheal lumen. Narrowing is most commonly caused by the presence of complete tracheal cartilage rings, as opposed to normal tracheal rings, which are U-shaped with posterior membranous septa [38,55]. The embryonic abnormality causing congenital tracheal stenosis is not known, although improper budding of the foregut may play a role [38]. Congenital tracheal stenosis is associated with cardiovascular abnormalities in as high as 50% of cases [55]. External compression from vascular anomalies can narrow the tracheal lumen, but unless there is an intrinsic airway anomaly, such as tracheal rings, the true tracheal caliber is normal and can be achieved by addressing the vascular anomaly. The incidence of congenital tracheal stenosis is 1 in 64,500 [56]. Infants with congenital tracheal stenosis present with dyspnea, stridor, recurrent respiratory tract infections, sudden collapse, and wheezing [57,58]. If the stenosis makes up less than 60% of the tracheal diameter, conservative treatment is appropriate [24]. In patients with more severe stenosis, surgical management is recommended. Slide tracheoplasty performed on extracorporeal membranous oxygenation is a highly specialized procedure that can provide a patent airway [59].

### 2.8. Craniofacial Anomalies

Multiple syndromes affect the facial and tongue anatomy, thereby compromising the neonatal airway.

#### 2.8.1. Pierre Robin Sequence

The Pierre Robin sequence, named after the eponymous French dental surgeon, is characterized by a clinical triad of micrognathia, glossoptosis, and airway obstruction. In most cases, a cleft palate is also present. Occurring in 1 in 8500–14,000 live births, PRS has an equal incidence between males and females, and can be isolated or part of a larger syndrome.

The tongue and mandible both originate from the mandibular arch, and their development is coordinated. Paranda and Chai note that the downward growth of the mandible permits the tongue to descend and the palatal shelves to reorient from vertical to horizontal. Still, no direct evidence exists demonstrating the causal relationship between mandibular malformation, tongue positioning, and cleft palate [60].

Due to the phenotypic heterogeneity of PRS, neonates can present with varying degrees of upper-airway obstruction. The obstruction in PRS is the result of posterior displacement of the tongue base, narrowing the oropharyngeal airway. Both physical exam and laryngoscopy can elucidate the extent of obstruction [61].

Depending on the severity of the obstruction, management can range from noninvasive maneuvers—including prone positioning and nasopharyngeal stenting—to surgery. The two most common surgeries to address airway obstruction in PRS are mandibular distraction osteogenesis (MDO) and tongue–lip adhesion (TLA). Both surgeries have been found to be effective in preventing a tracheostomy and restoring oral feeding ability. Patients who underwent MDO had significantly decreased need for secondary intervention for recurrent obstruction compared to those who received a TLA. However, patients who undergo MDO require an additional procedure for device removal and can experience complications including scarring, dental damage, and facial nerve injury [62]. Otherwise, specific craniofacial anomalies are beyond the scope of this review.

#### 2.8.2. Treacher Collins Syndrome

Treacher Collins Syndrome (TCS), a rare craniofacial syndrome occurring in roughly 1 out of 50,000 live births, is the result of abnormal differentiation of the first and second pharyngeal arches. Structures that form from these arches include the maxilla, zygomatic bone, mandible, a portion of the temporal bone, and the ossicles. Thus, patients with TRS can exhibit an array of phenotypes such as mandibulomaxillary hypoplasia, cleft palate, and conductive hearing loss (Figure 6). Although additional malformation such asperiorbital anomalies and microtia can occur, the main presenting features of TRS reflect malformation of the first two arches.

The Pierre Robin sequence, which causes airway obstruction as previously described, can occur in patients with Treacher Collins syndrome. Mandibular shape in TCS, however, is different from that of isolated PRS in that hypoplasia can occur in two axes, which can require multivector distraction. Like in PRS, prone or lateral positioning can assist neonatal ventilation. If these positions successfully reduce the obstruction, outpatient management is appropriate. If such maneuvers fail and the neonate requires additional support with oxygenation through positive pressure or intubation, additional inpatient workup should be performed, including polysomnography and laryngoscopy or bronchoscopy. In addition to identifying the level of airway obstruction, direct visualization can rule out additional anatomic causes of airway obstruction.

As in PRS, mandibular distraction osteogenesis and tongue–lip adhesion are common surgical interventions that begin in the first year of life to open the upper airway. However, if the airway compromise is too severe, a tracheostomy is unavoidable. In the past few years, there has been a new understanding into the anatomic complexity of the airway in patients with TCS. Ma and colleagues found that the TCS upper-airway volume was decreased by 30% compared to control, with the retroglossal airway being most affected. With respect to the nasal airway, the anterior-inferior portion of the nasal cavity decreased airway volume by 40%. These results suggest surgical intervention in TCS patients should extend beyond the mandible. In fact, midface surgery has been used to successfully decannulate four out of five patients who had previously undergone MDO in their series. These new surgical approaches are the direct result of our new understanding of the complex anatomy in TCS patients [63].

#### 2.8.3. Goldenhar Syndrome

Goldenhar syndrome was first described by Maurice Goldenhar in 1952, and the abnormalities she described include defects of structures derived from the first and second branchial arches. The severity of craniofacial deformities can range drastically among patients. Examples include hemifacial microsomia, mandibular and/or maxillary hypoplasia, and cleft lip and palate. The maxillary and mandibular retrusion, as well as facial bone hypoplasia, constrict the oropharyngeal airway and can result in significant obstruction, causing stridor, cyanosis, retractions, difficult intubation, and increased work of breathing. Airway difficulty can present in infants and worsen with age. Abnormalities of the pterygoid processes and adenoids, if present, can also affect the nasopharyngeal airway. Patients with airway symptoms should be evaluated and managed based on severity of symptoms. Treatment options include tonsillectomy and adenoidectomy, uvulopalatopharyngoplasty, turbinate reduction, or septoplasty. Tracheostomy may be necessary for severe obstruction. More complex procedures to correct facial bone deformation also provide significant benefit [64].

#### 2.8.4. Beckwith–Wiedeman Syndrome

Neonatal macroglossia is seen in several syndromes, including Beckwith–Wiedemann syndrome, Robinow Syndrome, and Down Syndrome. Patients with Beckwith–Wiedemann syndrome, referred to commonly as congenital overgrowth syndrome, have features such as macrosomia, macroglossia, midface hypoplasia, a prominent occiput, ear creases or pits, and nevus flammeus. Patients with Robinow syndrome can have macroglossia, ogival palate, absent or bifid uvula, ankyloglossia, and other craniofacial abnormalities. In both syndromes, occlusion of the oropharynx by the enlarged tongue may require partial glossectomy. Macroglossia is much more common in Beckwith–Wiedemann syndrome, in which the tongue grows slowly over the first year [65].

#### 2.8.5. Down Syndrome

Trisomy 21, or Down syndrome, is the most common chromosome abnormality in humans and occurs in 1 out of every 1000 births.

There are multiple anatomic causes of upper-airway obstruction for the Down syndrome patient, including midface hypoplasia, macroglossia, shortened palate, and narrow nasopharynx (Figure 7). Generalized hypotonia, a poor immune system, increased likelihood of GERD, and tendency to obesity further contribute to airway obstruction. The anatomy of the airways of children with Down Syndrome is smaller than that of an age-matched control subject, and in patients requiring intubation, a smaller endotracheal tube should be used to prevent airway trauma [66,67].

#### 2.8.6. Apert and Crouzon Syndrome

Patients with either Apert or Crouzon syndrome can also present with airway symptoms. Nearly all patients with Apert syndrome present with coronal craniosynostosis in addition to a hypoplastic midface. The airway of these patients can be obstructed at multiple locations—the nasal passages can be narrowed, the tongue can be withdrawn, and various tracheal anomalies can be present. Any patient with craniosynostosis should visit an ENT for airway monitoring and flexible laryngoscopy to assess sites of obstruction. For neonates with respiratory compromise, early intervention may be warranted [36].

As in most cases of neonatal airway compromise, the extent of management depends on symptoms and degree of obstruction. Neonates with mild symptoms should be managed conservatively. For example, if the only area of obstruction is choanal narrowing, nasal stents may be adequate. Those with more severe symptoms of apneic episodes or respiratory distress should be aggressively treated, including intubation or tracheostomy tube placement if appropriate [35,36].

Crouzon syndrome is the mildest of the craniosynostosis syndromes. Like in Apert Syndrome, the midface in Crouzon syndrome is retruded, although with less vertical impaction. However, in Crouzon’s, the midface is fully developed. Another difference is that cleft palate, which is common in Apert’s syndrome, is rarely found in Crouzon’s. Among the many facial anomalies present in Crouzon syndrome, a few can directly affect the airway, including septal deviation, nasal and choanal abnormalities, and nasopharyngeal narrowing. Often, obstruction is multilevel. Interestingly, the degree of obstruction may shift as the child grows [37].

### 2.9. Tracheal Agenesis

Tracheal agenesis is a rare condition where the trachea fails to develop normally. Occurring in roughly 1 in 50,000 newborns, it results from aberrant septation and separation of the lung buds from the foregut, leading either to complete absence or significant underdevelopment [68,69]. Tracheal agenesis can be classified into three types depending on the presence and position of the carina. Type 1 occurs when the carina is intact and the trachea inserts into the esophagus, type 2 when the carina itself inserts, and type 3 when the carina is absent and the bronchi insert directly [70,71].

Tracheal agenesis should be suspected in any neonate who is immediately hypoxemic and unable to be intubated [72]. A lack of audible cry in an otherwise responsive neonate may also indicate its presence [73]. Tracheostomy is viable only if sufficient tracheal tissue is available and does not provide a long-term solution. While several methods of reconstruction have been attempted, few have yielded promising results, rendering this condition largely fatal [73]. Creations of a neotracheal utilizing esophageal tissue in patients with existing tracheoesophageal fistula have been reported but should not be considered a standard of care [74].

### 2.10. Vallecular Cyst

Vallecular cysts are rare formations at the base of the tongue that can potentially obstruct the airway and interfere with feeding. They are thought to arise via submucosal duct obstruction in the vallecula or lingual epiglottis [75]. Presenting in neonates with inspiratory stridor and dyspnea most often within one week of birth, vallecular cysts cause varying degrees of airway obstruction in relation to their size [75]. Vallecular cysts alone comprise up to 2% of neonatal stridor, but also commonly present with laryngomalacia, contributing to the supraglottic collapse that confers airway compromise and reflux [75,76].

Diagnosis can be made via flexible endoscopy or laryngoscopy, and a high degree of suspicion is required to avoid misdiagnosing a patient with stridor. Direct laryngoscopy in the operating room may be required if the patient decompensates acutely or visualization is difficult [77]. The anomaly can be corrected by aspiration of the cyst and subsequent laser ablation, marsupialization, or micodebridement [75,78]. Roughly 15% of cysts recur, particularly when cysts invade through the cricothyroid membrane, but revision surgery has been shown to prevent further recurrence [75,79].

### 2.11. Congenital Neck Masses

Congenital neck masses pose a significant threat to airway patency, causing compression of the oropharynx, larynx, or trachea [80,81]. Masses are often diagnosed on prenatal ultrasound, but a significant portion go unnoticed until birth. Acute decompensation from any neck mass necessitates emergent intubation or tracheostomy if intubation is not possible. Presence of large masses or those that may cause airway compromise warrant consideration of delivery via ex utero intrapartum (EXIT) procedure, in which the neonate is partially delivered but maintained on maternal circulation until the airway is secured, sometimes including immediate tracheostomy.

Cervical lymphangiomas, also referred to as cystic hygromas depending on their size, make up a significant proportion of these masses, occurring in roughly 1 in 12,000 births [82]. Laryngeal involvement is rare, found in up to 6% of cases, but can compromise the airway if precipitating laryngeal edema [83,84]. Surgical removal is the definitive treatment, though initial mass reduction with sclerotherapy is often necessary [85]. Aspiration is not necessary unless the mass is acutely obstructing the airway [83].

Congenital teratomas also present a significant threat of airway compromise. Teratomas are a rare germ-cell tumor in children, occurring in 1 in 4000 births, and masses of the neck occur in up to 13% of these cases [81,86,87]. Oropharyngeal teratomas, also known as epignathi, and cervical teratomas are often resectable immediately after EXIT procedure, but diagnosis prenatally may be difficult, particularly in differentiating the teratoma from other common congenital masses [88,89]. Head and neck teratomas present a significant risk of airway obstruction that can result in fetal polyhydramnios and subsequent pulmonary hypoplasia, with incidence of postdelivery airway emergency as high as 35% (Figure 8) [90]. Thoracic teratomas have also been shown to cause tracheal compression, though their incidence is exceedingly rare [88]. Surgical resection is the definitive treatment, and while recurrence is rare, routine follow-up is necessary to monitor for malignant transformation or missed lesions [91].

Branchial cleft cysts (BCC) have potential to compress the airway. Remnants of embryologic structures in the lateral neck, BCCs make up 20% of all congenital neck masses [92]. While the majority of cysts are uncomplicated, causing intermittent drainage and swelling, extension of the cyst into the larynx can cause acute airway obstruction, particularly in the context of infection [92,93]. Cyst excision via an EXIT procedure or shortly after birth may be warranted depending on the size, as rapid or unpredictable cyst expansion may acutely compress the airway [94].

Some additional, rare neck masses that may compress the airway include plexiform neurofibromas, thyroglossal duct cysts, and lipoblastomas [95,96,97].

### 2.12. Nasal-Cavity Abnormalities

Generalized nasopharyngeal and oropharyngeal edema pose a significant risk of airway obstruction in neonates. Neonatal rhinitis is the most common cause of nasal obstruction in neonates, and may be precipitated by a number of factors, including birth trauma and inflammatory causes, though a significant proportion are idiopathic [98,99]. A higher incidence is often seen in colder weather or winter months [100]. Patients most often respond to medical treatment with saline and/or steroid drops, and persistent inflammation should be worked up for structural or immunological abnormalities such as choanal atresia [99]. As neonates are obligate nasal breathers, even a slight reduction in nasal-airway patency can predispose to respiratory distress. Neonatal adenoid hypertrophy may present early in life and precipitate oropharyngeal edema that obstructs the airway. While rare before 6 months of age, with most cases presenting after 18 months, adenoid hypertrophy can be produced by edema in neonates who exhibit severe reflux [101]. Optimal positioning and thickened feeds may assist in alleviating progression, though adenoidectomy is the definitive treatment [102].

#### 2.12.1. Congenital Nasal Pyriform Aperture Stenosis

Congenital nasal pyriform aperture stenosis (CNPAS) is a condition in which overgrowth of the maxillary nasal processes causes narrowing of the pyriform aperture, the anterior opening of the nasal cavity [103]. It is associated with central mega incisor and holoprosencephaly, as well as hypothalamopituitary axis abnormalities [104,105]. An association with maternal diabetes mellitus (MDM) has also been posited, with up to 55% of cases involving a mother with MDM [105]. Patients may present with noisy breathing, stridor, difficulty feeding, and respiratory distress [105,106]. Diagnosis can be made via CT scan, with an aperture diameter less than 11 mm indicating CNPAS (Figure 9) [107]. Conservative management with decongestants and mucus clearance may improve symptoms, but often need surgical correction depending on the severity of the stenosis [106]. A sublabial approach to the aperture with removal of the bony maxillary prominence is a common surgical treatment [105,106].

#### 2.12.2. Choanal Atresia

Choanal atresia is a narrowing or congenital closure of the choanae, or the posterior aperture of the nasal cavity into the nasopharynx. Found in 1 in every 5000–8000 children, it can present with either unilateral or bilateral closure of the choanae. While the specific embryological basis is unclear, the apertures are covered by a bony or mixed bony/membranous plate, and may copresent with other craniofacial abnormalities [108]. There may be concomitant atretic plate overgrowth extending into either aperture, as well as medialization of the lateral pterygoid plate [109,110]. Bilateral atresia presents at birth, with emergent respiratory distress and stridor. A distinctive feature is paradoxical cyanosis, wherein the patient resolves their cyanosis when crying, as air flow is temporarily reestablished [110,111]. Unilateral atresia normally presents later in life with symptoms of obstruction and unilateral rhinorrhea (see Figure 10), and may have several instances of otitis media in the ipsilateral ear [112]. Some patients are diagnosed after failure to pass a nasogastric tube or suction catheter. Diagnosis can be made with nasal endoscopy, though care must be taken not to enter the cranial cavity if other craniofacial abnormalities are present. CT definitively diagnoses the condition, but patients with bilateral atresia must be stabilized first via intubation or an oral airway [110]. Surgical correction can be achieved through transnasal puncture with dilators followed by temporary stents, especially in the case of mixed bony/membranous atresia, though a decrease in restenosis after stenting has not been demonstrated [113,114]. A transpalatal or endoscopic approach may also be taken to reduce the stenosis on the atretic and medial pterygoid plates, though rates of revision surgery up to 79% were found [108].

## 3. Conclusions

Neonatal airway anomalies can range from isolated aberrancies to those within a spectrum of multisystem involvement. Patients can present with minimal symptoms, such as noisy breathing, to severe respiratory distress requiring surgical airway. A high degree of suspicion is required to identify airway anomalies, and those treating neonates should be familiar with the range of anomalies described in this text.

## Figures and Tables

**Figure 1 children-09-00944-f001:**
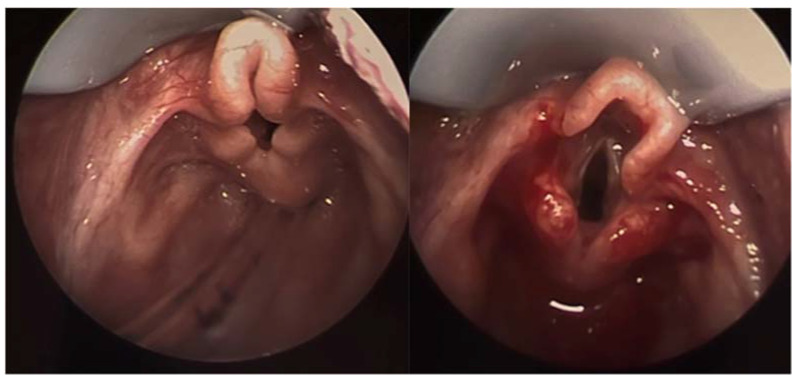
Laryngoscopy image showing significant laryngomalacia. Note the curled appearance of the airway with obstruction of the glottis. The image on right demonstrates the same patient immediately after supraglottoplasty.

**Figure 2 children-09-00944-f002:**
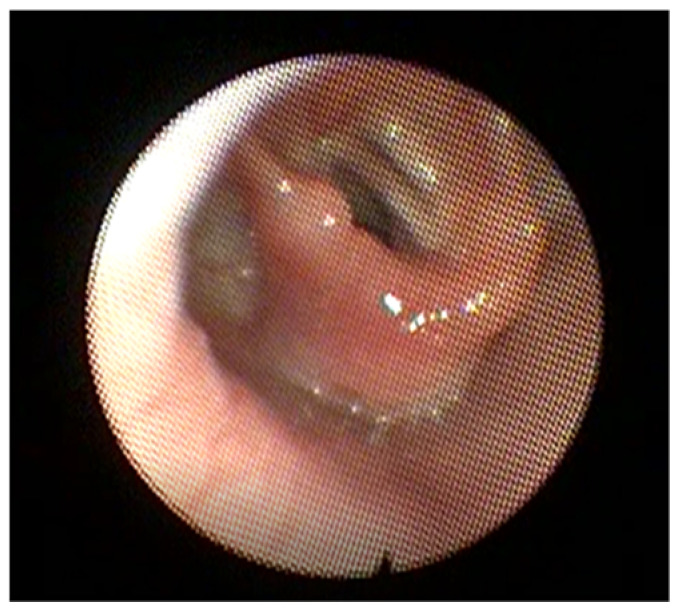
Flexible endoscopy image showing a unilateral vocal cord paralysis. The right vocal fold is abducted while the left is fixed in place.

**Figure 3 children-09-00944-f003:**
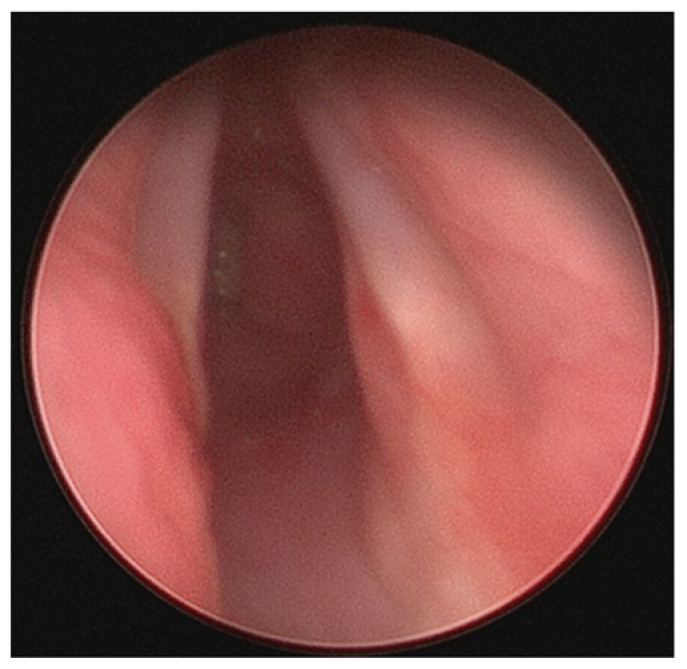
Severe subglottic stenosis.

**Figure 4 children-09-00944-f004:**
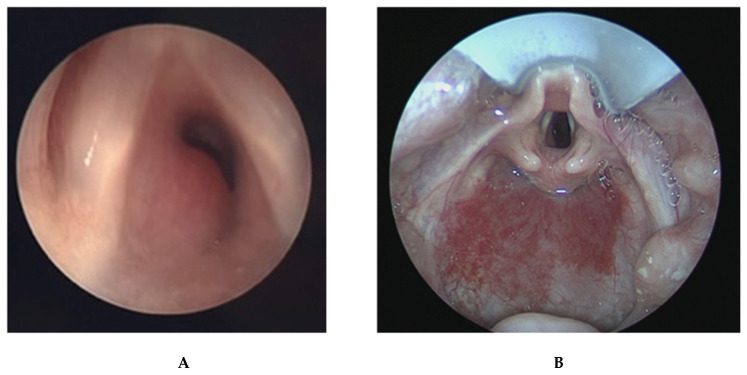
(**A**) Laryngoscopy image demonstrating large subglottic hemangioma obstructing a significant portion of subglottic airway. (**B**) This image demonstrates a more proximal view of the same patient’s airway showing improved subglottic patency after endoscopic treatment. Note the concomitant hypopharyngeal hemangioma.

**Figure 5 children-09-00944-f005:**
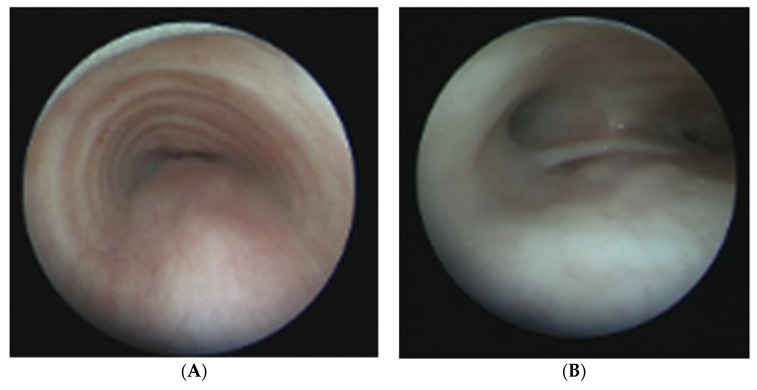
(**A**) Bronchoscopic images of an infant with severe tracheomalacia. (**B**) Bronchoscopic image of the distal trachea in an infant with severe TBM including a distal tracheoesopheal fistula.

**Figure 6 children-09-00944-f006:**
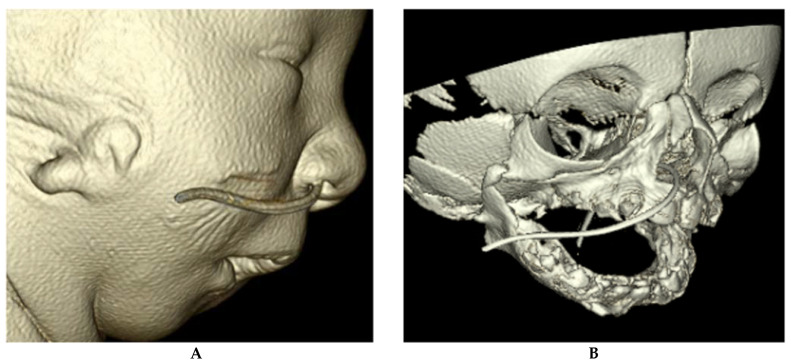
(**A**) CT reconstruction images of a patient with Treacher Collins syndrome. (**B**) CT reconstruction image of a patient with Treacher Collins syndrome. Note the maxillary and mandibular hypoplasia and micrognathia. This patient had bilateral microtia.

**Figure 7 children-09-00944-f007:**
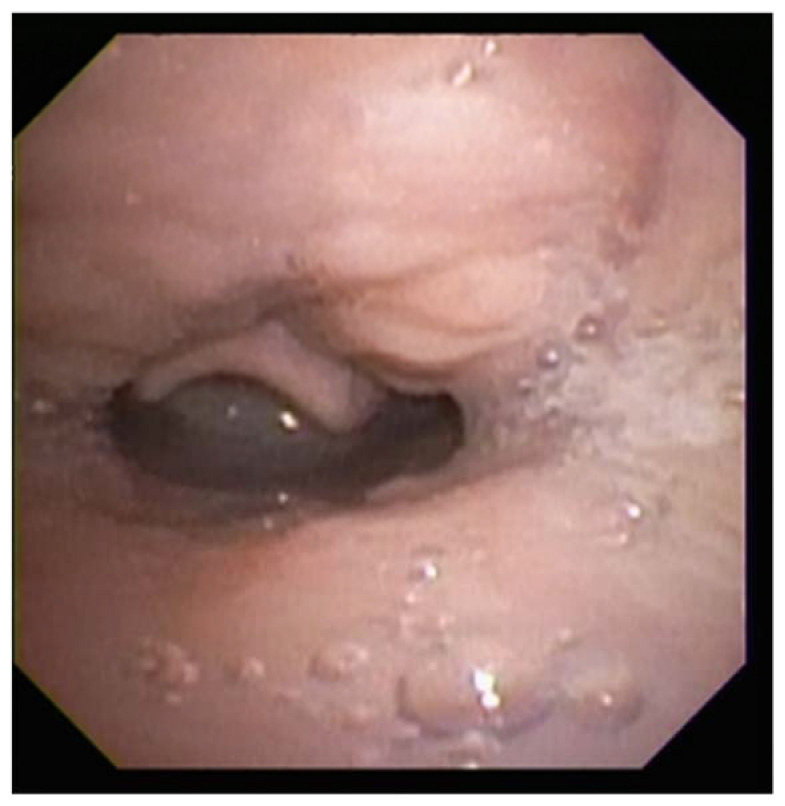
Sleep endoscopy image demonstrating tongue-base obstruction. Note the tongue base in contact with the epiglottis, which is prolapsed over the airway.

**Figure 8 children-09-00944-f008:**
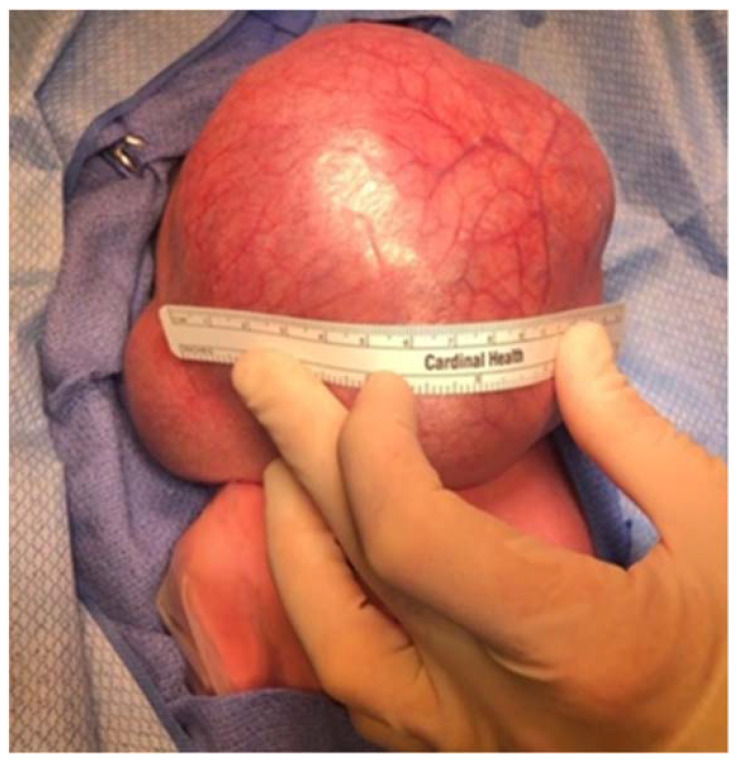
Infant with massive cervical teratoma. Infant was delivered via EXIT procedure with immediate mass resection following intubation while on maternal circulation.

**Figure 9 children-09-00944-f009:**
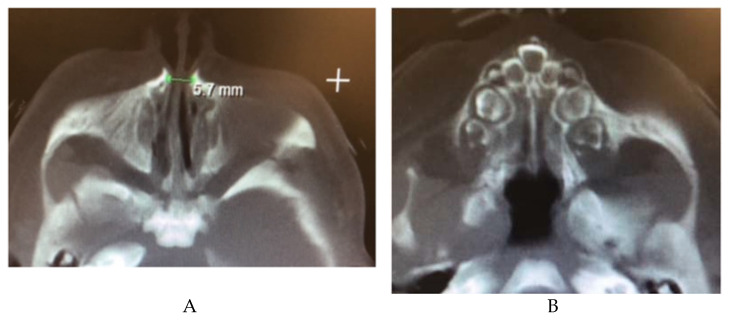
(**A**) CT image showing a narrow pyriform aperture (**B**) CT image showing central megaincisor.

**Figure 10 children-09-00944-f010:**
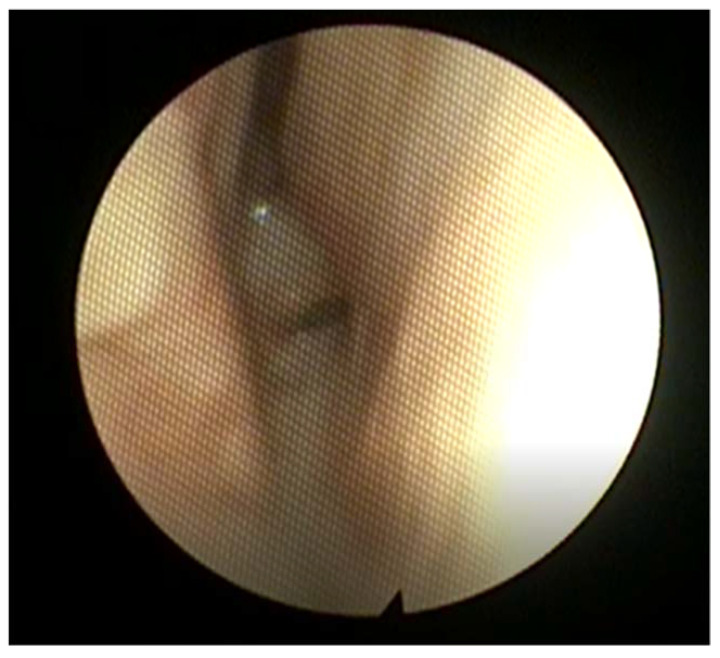
Posterior nasal endoscopy in a patient with severe choanal stenosis. Note the narrow choanal opening, with the septum to the right in this image.

## Data Availability

Not applicable.

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
