# Peer review of "Neonatal Airway Abnormalities"

_children, 2022, doi:10.3390/children9070944_

Round 1
Reviewer 1 Report
This is a very nice review of the subject.
Author Response
Appreciate comments. Thank you.
Reviewer 2 Report
Thank you for the opportunity to review this interesting manuscript describing Neonatal Airway Abnormalities. I would like to make following suggestions:
1) The manuscript is only a summary of various pathologies known to cause neonatal airway compromise. It does not add to the existing literature on this topic. Nor, does it elaborate on the most recent treatments,
2 2) Fig 4 : it would be interesting to see the corresponding subglottic endoscopic image that corresponds with prior treatment
Author Response
Thank you for your comments. We edited the text to include the most up to date treatment options for several of the conditions, including some very rare procedures that may be of interest to readers. We hope these updates, along with the extensive review, will allow dissemination of the text to a wider audience as it does provide a full overview of commonly and uncommonly encountered pediatric airway anomalies.
In regard to your comment on Fig 4, this was the imaging from the patient’s first bronchoscopy. Images from prior treatments were requested by the reviewer, but no such images exist. The only prior treatments for this patient were propranolol. The senior author would be happy to share additional clinical information or images within the text if useful.

Round 2
Reviewer 2 Report
Much improved version. Can be ACCEPTED